# The Effect of Apple Juice Concentration on Cider Fermentation and Properties of the Final Product

**DOI:** 10.3390/foods9101401

**Published:** 2020-10-02

**Authors:** Julia Rosend, Aleksei Kaleda, Rain Kuldjärv, Georg Arju, Ildar Nisamedtinov

**Affiliations:** 1Center of Food and Fermentation Technologies, Akadeemia tee 15A, 12618 Tallinn, Estonia; aleksei@tftak.eu (A.K.); rain@tftak.eu (R.K.); georg@tftak.eu (G.A.); 2Division of Food Technology, Department of Chemistry and Biotechnology, School of Science, Tallinn University of Technology, Ehitajate tee 5, 12616 Tallinn, Estonia; 3Division of Chemistry, Department of Chemistry and Biotechnology, School of Science, Tallinn University of Technology, Ehitajate tee 5, 12616 Tallinn, Estonia; 4Lallemand Inc., 1620 Rue Préfontaine, Montréal, QC H1W 2N8, Canada; inisamedtinov@lallemand.com

**Keywords:** cider, juice concentrate, fermentation, gas chromatography, volatile esters, hydrogen sulfide

## Abstract

European legislation overall agrees that apple juice concentrate is allowed to be used to some extent in cider production. However, no comprehensive research is available to date on the differences in suitability for fermentation between fresh apple juice and that of reconstituted apple juice concentrate. This study aimed to apply freshly pressed juice and juice concentrate made from the same apple cultivar as a substrate for cider fermentation. Differences in yeast performance in terms of fermentation kinetics and consumption of nutrients have been assessed. Fermented ciders were compared according to volatile ester composition and off-flavor formation related to hydrogen sulfide. Based on the results, in the samples fermented with the concentrate, the yeasts consumed less fructose. The formation of long-chain fatty acid esters increased with the use of reconstituted juice concentrate while the differences in off-flavor formation could not be determined. Overall, the use of the concentrate can be considered efficient enough for the purpose of cider fermentation. However, some nutritional supplementation might be required to support the vitality of yeast.

## 1. Introduction

Cider is a beverage produced by fermenting apple juice. The juice for the fermentation can be obtained in two different ways—either by having the juice freshly pressed and clarified or by reconstituting a juice concentrate to desired properties. The use of juice concentrates in the cider industry possesses several economic advantages, especially where mass production is concerned. Concentrates are available in large quantities, easily transportable (stable), and more affordable when purchased in bulk. However, freshly pressed apple juice allows more options due to blending juices from different apple varieties whereas the choice between different available apple cultivar concentrates is limited. 

The specific definition for cider is country dependent and also legislation on cider differs—some countries have strict legislation and cider is well defined, whereas other countries are still developing specific legislation for cider. In Estonia, cider can be prepared from apple juice obtained from freshly pressed apples or apple juice concentrate with no particular limit on the percentage of apple components [1]. On the other hand, for example, French cider production policy declares that the product named “cider” must derive from the fermentation of fresh apple juice or a mixture of several juices. In this instance, apple juice may be partially obtained from the concentrate only if the latter does not exceed 50% by volume [2]. In the UK, the pre-fermentation mixture should contain at least 35% apple juice [3]. 

The number of studies on cider properties and their development during fermentation has recently increased to a considerable extent. Most of these studies, however, have been largely focused on fermentation management aspects—the use of different apple varieties, ripening stage, yeast strains, and nitrogen supplementation [4,5,6,7,8,9,10,11,12,13,14,15,16,17,18,19]. There are no studies available that would directly compare freshly pressed apple juice and the concentrate made from the same juice pressing batch. This kind of research could potentially help to evaluate the effect of the juice concentration process on the fermentation and properties of the final product. Despite the economic advantages of the use of commercial apple juice concentrates in the cider industry, the definitive research on the differences in the performance is yet to be properly explored. 

The purpose of this research was to provide a comparison of fresh apple juice and concentrate as a substrate for cider fermentation based on yeast performance and properties of fermented cider in terms of fermentation kinetics, volatile ester production, and off-flavor development potential. 

## 2. Materials and Methods 

### 2.1. Apple Juice Clarification and Concentration 

This study used apple juice (Brix 9.80%; pH 3.38; titratable acidity 3.93 g L^−1^ in malic acid equivalent) industrially pressed from ripe Antei apples (150 L). Half of the juice (75 L) was clarified and concentrated. Clarification was based on the method described by Grampp et al. (1978) [20]. The juice was warmed to 40 °C from the storage temperature; 5 g hL^−1^ of commercial pectinase (Rapidase; Oenobrands SAS, Montferrier-sur-Lez, France) and 5 g hL^−1^ of glucoamylase (AMG 300L; Novozymes, Copenhagen, Denmark) were added. After 30 min at 40 °C, 100 mL hL^−1^ of silica sol (Baykisol^®^ 30; E. Begerow GmbH & Co., Langenlonsheim, Germany) and 10 g hL^−1^ of dissolved gelatin (SIHA Clarifying Gelatine; E. Begerow GmbH & Co., Langenlonsheim, Germany) were added as flocculants. Flocculation occurred within 1.5 h at 40 °C. Clarified juice was filtered and concentrated using a vacuum concentration system (Didacta Italy; Fruit juice and syrup line, code 640,022). Due to volume restrains, the process was performed in three batches of 25 L. Each batch was concentrated separately from 9.8% to 30% Brix. The batches were then pooled and concentrated to the final Brix value of 67.8%. For fermentation, the concentrate was diluted with distilled water to 10% Brix (pH 3.45; titratable acidity 3.97 g L^−1^ in malic acid equivalents).

### 2.2. Free Amino Acid Content 

Free amino acids were measured in the juice and reconstituted concentrate by UPLC (Acquity UPLC; Waters Corp., Milford, MA, USA) equipped with AccQ·Tag Ultra column and a UV detector according to the method applied by Lahtvee et al. (2014) [21]. Before analysis, the samples were diluted 1:2 with MilliQ, filtered (Whatman Spartan 13; Dassel, Germany), and derivatized using AccQ·Tag reagent. AccQ·Tag Ultra eluent B (linear gradient from 0% to 100%) was used as mobile phase at a flow rate of 0.3 mL min^−1^. Three analytical replicates were measured for each sample. 

### 2.3. Cider Fermentation 

Apple juice and reconstituted apple juice concentrate (400 mL) were distributed into sterile 500 mL Duran bottles (DWK Life Sciences; Mainz, Germany). Each bottle was inoculated (5 × 10^6^ CFU mL^−1^) with a chosen yeast starter culture (Lallemand, Inc.; Quebec, QC, Canada). The yeast starter cultures used in the study were as follows: Y1 (*S. bayanus*; red and white wine yeast for demanding conditions), Y2 (*S. cerevisiae*; white wine yeast), Y3 (*S. bayanus*; sparkling wine yeast), Y4 (*S. cerevisiae*; red, rosé, and white wine yeast, selected through evolutionary adaptation), Y5 (*S. cerevisiae* with killer factor), Y6 (*S. cerevisiae*; white wine yeast selected through directed breeding). The fermentation vessels were sealed using screw caps with septums pierced with a syringe needle (20 G × 1”, 0.9 × 25 mm; Terumo Medical Corporation, Somerset, NJ, USA) coupled with a microfilter (Millex PTFE Vent filters; Merck, Darmstadt, Germany) to vent carbon dioxide. In the headspace of each vessel, a piece of lead acetate paper strip (Whatman; GE Healthcare, Chicago, IL, USA) was placed to monitor hydrogen sulfide formation. The changes in color were assessed visually and grouped in comparison to blank paper strip and relatively to each other on a scale from “no color change” to “high-intensity color formation”. It needs to be emphasized that the use of lead acetate paper strips has certain limitations. The change in color intensity of the strip as a function of hydrogen sulfide content in the headspace is not linear which makes even relative quantification impossible—the strips become saturated after which no further change in color can be observed despite hydrogen sulfide still potentially being produced by the yeast. Three nutritional strategies were applied: no additional nutritional supplementation (control); 9 g hL^−1^ of diammonium phosphate (DAP; Sigma Aldrich, St. Louis, MO, USA) at the start of fermentation as inorganic nitrogen supplement; 40 g hL^−1^ of organic supplement Fermaid O (Lallemand Inc, Montreal, QC, Canada). Fermentations were carried out at 18 ± 1 °C and 30 ± 1 °C by following carbon dioxide dissipation (mass loss) every 24 h. Fermentations were completed when the weight of fermentation vessels remained constant for three consecutive days. Experiments were conducted in duplicates. The total number of fermentations was thus 144 (2 fermentation matrices × 6 yeasts × 2 temperatures × 3 nutritional strategies × 2 biological replicates). 

### 2.4. Fructose and Malic acid Content 

Fructose and malic acid content before and after fermentation were analyzed using HPLC (Alliance HPLC; Waters Corp., Milford, MA, USA), BioRad HPX87H column, RI and UV detectors. Prior to analysis, the samples were diluted 1:10 with MilliQ and filtered (Whatman Spartan 13; Dassel, Germany). 5.0 mM H_2_SO_4_ solution was used as mobile phase with a flow rate of 0.6 mL min^−1^. Standard solutions of fructose and malic acid were used for calibration curves. Two analytical replicates were measured for each sample. 

### 2.5. Analysis of Volatile Esters 

The presence of ethyl esters and acetate esters with a primary role in the formation and perception of fruity notes was monitored across the samples. The esters included in the study were as follows: ethyl acetate, isoamyl acetate, hexyl acetate, ethyl butanoate, ethyl hexanoate, ethyl octanoate, ethyl decanoate, and ethyl dodecanoate [15,22]. Quantification of volatile esters was performed according to the method previously described by Rosend et al. (2019) [15].

The samples were diluted 1:19 with distilled water into a 20 mL vial; 2-chloro-6-methylphenol (100 µg L^−1^) was added as an internal standard (IS) for quantification. Volatile compounds were extracted by using solid-phase microextraction (DVB/Car/PDMS 30/50 μm Stableflex, 2 cm; Supelco, Bellefonte, PA, USA) at 45 °C for 20 min. Quantification of volatile compounds was performed using a gas chromatograph system (6890N; Agilent Technologies, Santa Clara, CA, USA) equipped with a mass spectrometer (GCT Premier TOF; Waters, Milford, MA, USA) and a DB5-MS column (30 m × 0.25 mm × 1.0 μm; J&W Scientific, Folsom, CA, USA). Helium was used as a carrier gas with a flow rate of 1.0 mL min^−1^. The oven was programmed to ramp up from 40 °C at a rate of 7.5 °C min^−1^ to a final temperature of 280 °C with an additional holding time of three minutes (total run time 35 min). Mass spectra were obtained at ionization energy of 70 eV and a scan speed of 10 scans s^−1^, with a mass scan range of 35 to 350 Da. Three analytical replicates were used for each sample. Analytical standards were used for the accurate identification of selected compounds. The concentrations were expressed as µg L^−1^ in the internal standard equivalent. 

### 2.6. Sensory Assessment of Hydrogen Sulfide Related Off-Flavor 

In this study, a sensory panel consisting of 8 trained assessors with previous experience in cider assessment carried out the sensory analysis. Prior to the analysis, the panel was familiarized with hydrogen sulfide related off-flavor by using spiked reference samples of different intensities. The vocabulary was established for the description of the off-flavor (“rotten egg,” “cabbage,” “sulfuric”). The assessment scale was established as a 4-point category scale: “no perceived off-flavor,” “low-intensity off-flavor,” “moderate-intensity off-flavor,” and “high-intensity off-flavor.” 

All samples were encoded with a randomized three-digit number. The samples were served in sniffing glasses and presented in sequential monadic order. Two analytical replicates were used for each sample. Prior to sensory analysis, all cider samples were adjusted for sweetness to balance out the sourness since the secondary malolactic fermentation was not carried out. In the course of the analysis, the assessors were asked to assess the intensity of perceived off-flavor (cumulative off-flavor in odor and taste) on the established scale. Samples were assessed independently by each assessor. The results were taken as averages across all replicates and assessors. 

### 2.7. Data Processing 

The results of chemical analysis were averaged across biological and analytical replicates. The analysis of variance was performed using R version 4.0.0 (R Foundation for Statistical Computing, Vienna, Austria) and *p* < 0.05 was considered statistically significant. The results of GC-MS analysis were evaluated by partial least squares discriminant analysis (PLS-DA) using the R package ‘mixOmics’ 6.11.33 and presented as biplots. The response variable was constructed from a combination of juice type and nutritional supplement. Prior to the application of PLS-DA, the quantitation results were autoscaled. 

## 3. Results and Discussion

### 3.1. Effect of Juice Concentration on Free Amino Acid Composition 

The concentration of individual free amino acids (FAA) was measured in the juice and reconstituted apple juice concentrate prior to the start of fermentation and free amino nitrogen content (FAN) content was calculated (Table 1). Knowing the amount of added nutritional supplementation and FAN/NH_4_^+^ contained therein, the approximate amount of yeast assimilable nitrogen (YAN) in supplemented samples at the start of fermentation was also determined. 

The most abundant free amino acids in the apple juice used in the study were asparagine, aspartic acid, and glutamic acid. This correlates with what has been shown previously in other studies [23,24]. Overall statistical significance according to the analysis of variance showed that the juice and reconstituted concentrate can be considered as different (*p* < 0.05), the latter containing much more free amino acids. Thus, the concentration of serine, glycine, threonine, ornithine, lysine, tyrosine, isoleucine, and leucine was deemed statistically significantly higher (*p* < 0.05) and, hence, attributed to the process of juice concentration. The industrial concentration of juices is a high-pressure boiling process that has been reported to possess several adverse effects on the juice. One of them is thermal degradation and evaporation of volatile compounds which may increase the proportion of free amino acids [25,26,27,28]. The increase could also be attributed to partial peptidolysis/proteolysis that may have led to the release of individual amino acids. Finally, the increase of free amino acids concentration could also be attributed to a shift in proportions of the components of the juice matrix during reconstitution–the loss of some intrinsic constituents during clarification pre-treatment shifts the proportions in favor of amino acids when the water is added back according to % Brix. 

### 3.2. Fermentation Kinetics 

The fermentation kinetics results of ciders prepared from either fresh juice or juice concentrate, fermented with different yeast at two different temperatures with or without a nutrient are provided in Figure 1. In all samples, fermentations were successfully completed with no signs of sluggish fermentation.

Expectedly, some differences between strains could be observed while the temperature was the main factor affecting yeast fermentative activity. The yeast fermented at least two times faster at 30 °C. No particular difference in fermentative activity can be noted between the juice and reconstituted concentrate at 18 °C. However, at 30 °C, the fermentation activity on reconstituted concentrate was slightly inferior to the juice. A loss of certain nutrients during the concentration process might have affected the ability of yeast to cope with higher temperatures but the change is not drastic enough to significantly impact the effectiveness of the process. In this case, the supplementation fills the necessary nutritional gap enough to make the yeast perform better. 

### 3.3. Assimilation of Fructose 

Apple juice and its concentrate contained 8.08 ± 0.42% and 8.33 ± 0.26% of fructose before the start of fermentation, respectively. The amount of residual fructose in cider samples fermented under different conditions is shown in Figure 2. Based on the pattern of fructose consumption the yeasts used in the study can be divided into fructophobic and fructophilic (*p* < 0.05). The similar behavior in yeasts has been previously shown by Rosend et al. (2019) [15].

The yeasts left behind larger amounts of residual fructose in reconstituted concentrate (up to 1.29% residual fructose) in comparison to the fermentations performed with apple juice (up to 0.32% residual fructose). This observation correlates with the differences in fermentation kinetics shown previously (Figure 1). Fructophilic yeast (Y3, Y4) consumed the same amount of fructose regardless of temperature or applied nutritional supplementation. On the other hand, fructose assimilation by fructophobic yeasts (Y1, Y2, Y5, Y6) was slightly improved with nutrient addition, with Fermaid O being more effective than DAP (*p* < 0.05). Higher fermentation temperature was also shown to have a stimulating effect on fructose assimilation by generally fructophobic yeast used in this study. However, it should be emphasized that this might not be the case for other yeast species. For example, low-temperature fermentations have been previously reported as preferred for non-*Saccharomyces* yeasts [29,30,31].

### 3.4. Assimilation of Malic Acid

At the start of fermentation, apple juice contained 32.07 ± 2.24 mM of malic acid while the reconstituted concentrate 56.68 ± 4.47 mM. The higher malic acid content in the concentrate was most likely caused by removal/loss in other constitutes contributing to the Brix value during clarification and evaporation. 

The concentration of residual malic acid in cider samples fermented under different conditions in comparison to initial amounts are shown in Figure 3. In general, malic acid consumption by the yeast was rather low. Temperature and nutritional supplementation had no stimulating effect on malic acid assimilation. The main difference in malic acid consumption can be noticed only when comparing the juice and reconstituted concentrate (*p* < 0.05). Thus, malic acid consumption was higher in the environment with higher initial malic acid content. This can be tied to previous reports of a more efficient enzymatic conversion of malic acid to pyruvate at higher extracellular malate concentrations [15,32,33]. The K_m_ of Mae1p enzyme responsible for malic acid decarboxylation is reported at 50 mM which is close to the concentration of malic acid in juice concentrate [32].

### 3.5. Production of Volatile Esters

The presence of 8 esters was monitored across the samples. PLS-DA was employed as a statistical approach to observe the differences in ester production (Figure 4). PLS-DA method identifies the differences between the sample groups, which are shown as ellipses on the biplots with overlap indicating similarity. The characteristic differences between the groups are evaluated on an axis that goes through the origin point at (0,0) towards the position of a volatile ester label. The best representation of the results was achieved when viewing the samples according to the yeast species—*S. bayanus* yeasts (Figure 4A) were compared to *S. cerevisiae* yeasts (Figure 4B). According to the biplots, all ciders fermented with *S. cerevisiae* yeasts clustered closely together showing more consistency across the samples in volatile ester composition than *S. bayanus* yeasts.

The cider samples made with the concentrate correlated with the highest production of isoamyl acetate (up to 6700 µg L^−1^ in the concentrate; up to 5100 µg L^−1^ in the juice). Isoamyl acetate can be synthesized by yeast either from amino acids leucine and valine or from isoamyl alcohol [34,35]. As was noted previously, the diluted concentrate had elevated amounts of most of the identified amino acids, including leucine. Higher content of leucine might have been successfully utilized by the strains used in this study for isoamyl acetate production. *S. cerevisiae* yeasts used in the study accounted for the highest amount of isoamyl acetate produced (up to 6700 µg L^−1^) as a result of additional leucine utilization than *S. bayanus* yeasts (up to 5300 µg L^−1^). 

According to the biplots, the use of juice concentrate in cider fermentation has also resulted in the production of selected long-chain fatty acid esters (ethyl decanoate, ethyl dodecanoate). Long-chain fatty acid esters possess a characteristic “fatty” or “soapy” odor description; their overproduction when using juice concentrate in fermentations might increase the risk of the off-flavor formation [36,37]. On the other hand, the same esters have also been reported to increase the perceived fruitiness in wine [38]. 

It should be noted that long-chain fatty acids esters have been previously tied to the onset of cell death [39,40,41]. Due to their low permeability through the cell membrane, the long-chain fatty acid esters are released along with other compounds (e.g., amino acids, fatty acids, lipids, glycoproteins, mannoproteins) into the environment during cell autolysis [39]. The correlation of long-chain fatty acids esters with the cider samples produced using concentrate could signify a higher cell death rate in comparison to the samples prepared with apple juice. 

The nutritional supplementation approaches used in this study did not possess any significant influence on the production of volatile esters in *S. ceverisiae* strains. On the other hand, organic supplementation has increased the production of medium-chain fatty acid ethyl esters (ethyl butyrate, ethyl hexanoate, ethyl octanoate) in *S. bayanus* strains and reduced the formation of long-chain fatty acid esters. Supplementation of nitrogen with organic sources has proven to be efficient at stimulating ethyl ester production in multiple previous studies [7,38,42]. The supplementation requirements for the visible increase in ester production, however, have shown in this study to be strain-specific. 

### 3.6. Hydrogen Sulfide Production and Off-Flavor Development

Hydrogen sulfide production in the cider samples was assessed by using indicative lead acetate paper strips in the headspace of fermentation vessels. Due to the limitations of the method, conclusive quantitative decisions on the differences in hydrogen production cannot be made. Hence, the conclusions are provided as estimates only. 

The H_2_S production determined by the color intensities of the reaction on the lead acetate paper strip and the respective intensities of the sensorially perceived sulfuric off-flavor are provided in Table 2 and Table 3, respectively. The pattern of hydrogen sulfide production was strain-specific, which has also been noticed in other studies [43,44]. Most of the strains used in this study produced hydrogen sulfide during fermentation but generally not enough to produce sulfuric off-flavor of high intensity. Y6 (*S. cerevisiae* strain) has been declared by a manufacturer as a yeast with a low capacity to produce hydrogen sulfide. According to the results of our experiments, it indeed showed the lowest production of hydrogen sulfide. 

In general, lower temperature fermentation (18 °C) resulted in higher production of hydrogen sulfide in the headspace by the end of the fermentation. This can be attributed to the better accumulation of the hydrogen sulfide due to low fermentation speed and prolonged viability of the yeast culture [45]. No notable differences in hydrogen sulfide production (reaction on the test strip) could be observed between apple juice and concentrate. The addition of nutrients prior to fermentation did not reduce hydrogen sulfide formation and, in some cases, even promoted it. For example, hydrogen sulfide production increased with the addition of a nutritional supplement in ciders fermented with Y5 yeast. The increase, however, was intense enough to produce a perceived off-flavor only in the case of DAP addition.

## 4. Conclusions

Industrially prepared apple juice concentrate was successfully applied in this study to observe the differences in cider fermentation in comparison to fresh apple juice. The process of clarification and concentration was shown to affect the concentration of initial amino acids and malic acid—both increased significantly after the treatment. In terms of fermentation kinetics, the concentrate was shown to be slightly inferior at higher temperatures to the fresh apple juice, most likely due to the partial loss of nutrients. Such was also evident from the fructose consumption patterns. As expected, the increased malic acid concentration in the concentrate increased also its consumption by yeast. The production of volatile esters was also affected by the use of the concentrate. When using the concentrate all yeast strains under study showed increased production of long-chain fatty acid esters (ethyl decanoate, ethyl dodecanoate) which might signify a higher cell death rate in the cider samples fermented with the concentrate. Increased synthesis of isoamyl acetate was also noted in the samples fermented with the concentrate which could be attributed to the higher concentration of leucine in it, which is a precursor for isoamyl acetate synthesis. This effect was, however, specific to the species of the yeast used as was noticed mainly with *S. cerevisiae* strains. No differences in the hydrogen sulfide related off-flavor formation was observed between the fresh apple juice and the concentrate. In conclusion, the use of the apple juice concentrate can result in rather similar cider fermentation kinetics and quality as in the case with fresh apple juice; however, the fermentation might require additional nutritional supplementation to compensate for the loss of some nutrients and support the viability of the yeast cell.

## Figures and Tables

**Figure 1 foods-09-01401-f001:**
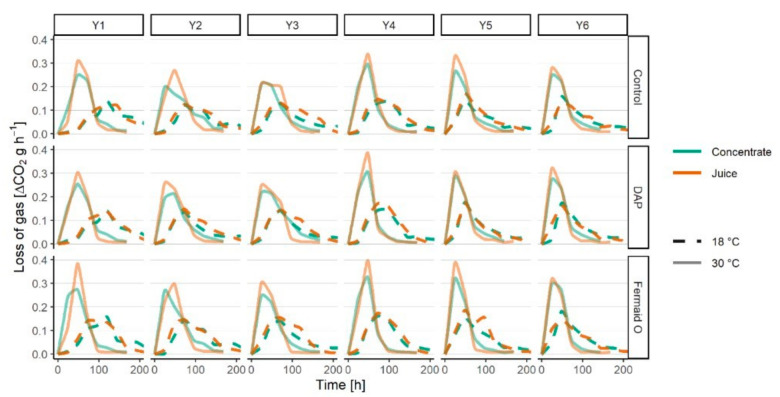
Fermentation kinetics of cider samples prepared from fresh juice or juice concentrate with different yeasts (Y1-Y6) at 18 or 30 °C. The samples done with the concentrate are represented by a green line and with the juice by an orange line. The samples fermented at 18 °C are represented by a dashed line of appropriate color; by a solid line at 30 °C.

**Figure 2 foods-09-01401-f002:**
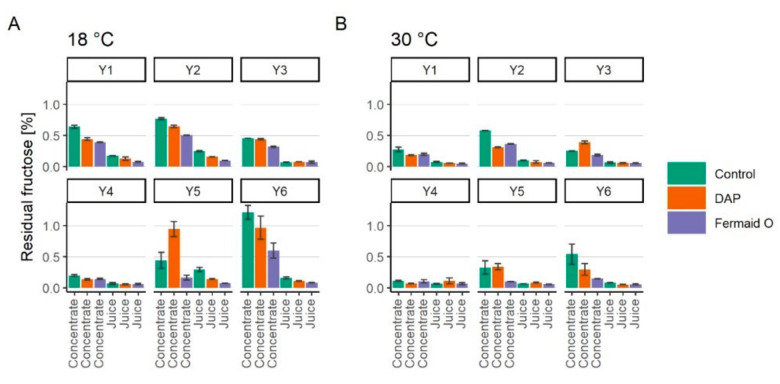
Residual fructose concentration at the end of fermentations prepared from fresh juice or juice concentrate with different yeasts (Y1-Y6) at 18 or 30 °C (*p* < 0.05). The color green represents control samples, orange—supplemented with DAP (diammonium phosphate), purple—supplemented with Fermaid O. (**A**) represents the samples fermented at 18 °C; (**B**)—at 30 °C. Red lines represent initial malic acid content either in juice or reconstituted concentrate.

**Figure 3 foods-09-01401-f003:**
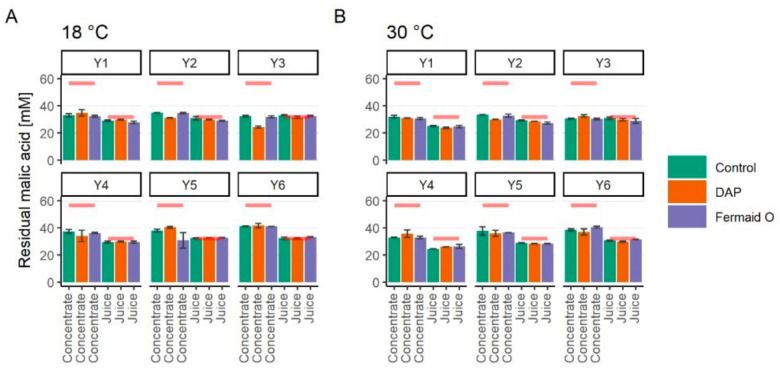
Residual malic acid concentration at the end of fermentation (*p* < 0.05). The color green represents control samples, orange—supplemented with DAP, purple—supplemented with Fermaid O. (**A**) represents the samples fermented at 18 °C; (**B**)—at 30 °C. Red lines represent initial malic acid content either in juice or reconstituted concentrate.

**Figure 4 foods-09-01401-f004:**
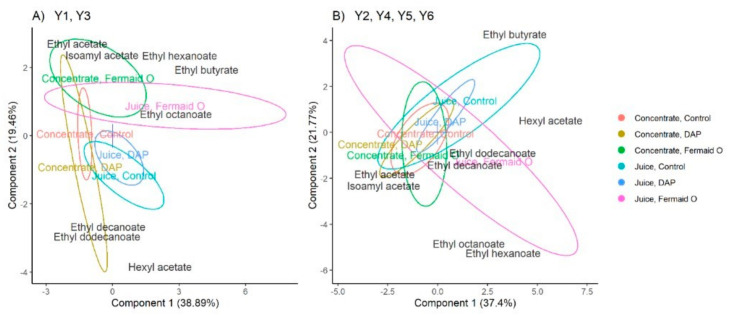
PLS-DA biplots of volatile ester production in cider samples. (**A**) *S. bayanus* yeasts; (**B**) *S. cerevisiae* yeasts. Colors correspond to the combinations of juice type and nutritional supplement. Ellipses indicate confidence region at 0.95 level. The number in parenthesis shows variance captured by the component.

**Table 1 foods-09-01401-t001:** Free amino acid (FAA) composition, free amino nitrogen (FAN) content before and yeast assimilable nitrogen (YAN) content after supplementation. Standard deviation is shown (*n* = 3).

FAA, mg L^−1^	Juice, 9.8% Brix	Concentrate, 10% Brix
His	1.71 ± 0.32	2.27 ± 0.44
Asn	269.60 ± 24.48	315.03 ± 32.70
Ser	6.28 ± 0.93	9.21 ± 0.82
Gln	3.26 ± 0.60	2.87 ± 0.26
Arg	21.68 ± 1.10	19.36 ± 3.63
Gly	0.32 ± 0.04	1.19 ± 0.21
Asp	38.10 ± 5.26	44.42 ± 4.46
Glu	32.43 ± 3.54	38.15 ± 3.67
Thr	2.75 ± 0.45	3.62 ± 0.33
Ala	8.85 ± 1.56	10.91 ± 1.01
Pro	2.32 ± 0.37	3.22 ± 0.33
Orn	0.27 ± 0.02	0.51 ± 0.10
Cys-cys	0.30 ± 0.03	0.34 ± 0.03
Lys	0.82 ± 0.09	1.93 ± 0.08
Tyr	0.88 ± 0.06	1.04 ± 0.22
Met	1.06 ± 0.16	1.27 ± 0.12
Val	3.93 ± 0.73	4.41 ± 0.46
Ile	2.34 ± 0.45	2.90 ± 0.33
Leu	1.18 ± 0.23	3.21 ± 0.39
Phe	1.97 ± 0.16	1.98 ± 0.21
Trp	0.57 ± 0.10	0.41 ± 0.03
FAN, mg L^−1^	76.13 ± 7.29	87.91 ± 9.62
+ DAP YAN, mg L^−1^	95.03 ± 7.29 ^1^	106.81 ± 9.62 ^1^
+ Fermaid O YAN, mg L^−1^	93.33 ± 7.29 ^2^	105.11 ± 9.62 ^2^

^1^ Taking into consideration that DAP contains 21% on nitrogen by mass. ^2^ According to the manufacturer’s instructions, 40 g hL^−1^ dose of Fermaid O is equivalent to 17.2 mg L^−1^ of YAN.

**Table 2 foods-09-01401-t002:** Color reaction intensity of H_2_S (hydrogen sulfide) formation on the selective lead acetate paper strip during fermentations prepared from fresh juice or juice concentrate with different yeasts (Y1-Y6) at 18 or 30 °C. − No color change; + Low intensity color formation; ++ Moderate intensity color formation; +++ High intensity color formation.

Nutrient	Control				Fermaid O				DAP *			
Matrix	Juice		Concentrate		Juice		Concentrate		Juice		Concentrate	
°C	18	30	18	30	18	30	18	30	18	30	18	30
Y1	++	−	++	++	+++	++	++	++	+++	++	++	++
Y2	++	−	++	−	++	−	++	++	++	−	++	+
Y3	++	++	++	++	++	+	++	++	++	+	++	+
Y4	+++	−	+++	+	+++	−	+++	++	+++	−	+++	+++
Y5	+	+	+	+	++	++	++	++	+++	+++	+++	+++
Y6	−	−	−	−	+	−	+	−	−	−	−	−

* Diammonium phosphate.

**Table 3 foods-09-01401-t003:** Sulfur off-flavor occurrence and its intensity in cider samples. **−** No off-flavor perceived; + Low intensity off-flavor; ++ Moderate intensity off-flavor; +++ High intensity off-flavor.

Nutrient	Control				Fermaid O				DAP			
Matrix	Juice		Concentrate		Juice		Concentrate		Juice		Concentrate	
°C	18	30	18	30	18	30	18	30	18	30	18	30
Y1	+	−	−	−	++	−	−	−	++	−	−	−
Y2	−	−	−	−	−	−	−	−	−	−	−	−
Y3	−	−	−	−	−	−	−	−	−	−	−	−
Y4	+	−	++	−	+	−	++	−	++	−	++	++
Y5	−	−	−	−	−	−	−	−	++	++	++	++
Y6	−	−	−	−	−	−	−	−	−	−	−	−

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
