# Peer review of "The Effect of Apple Juice Concentration on Cider Fermentation and Properties of the Final Product"

_foods, 2020, doi:10.3390/foods9101401_

Round 1

Reviewer 1 Report

The overall quality of the manuscript is fair. However some remarques should be explained and improved by the authors:

Lines 135-136: “The assessment scale was established as a category scale ranging from “no perceived off-135 flavor" to “high intensity off-flavor". The category scale present how many “categories” or “different intensities positions”? This information should be provided.

Why only sulphur related compounds were used in training of the sensory panel? In lines 256 and 257, authors refer that “… Long chain fatty acid esters possess a characteristic “fatty” or “soapy” odor description;” These descriptors should also be evaluated in training sessions.

Lines 276-282: Some information of this sentences should be removed and placed in Materials and Methods section. Moreover, the trademark and supplier of the lead acetate paper strips should be provided. Authors should explain why they didn't used a more accurate and reliable method in order to quantify hydrogen sulfide. In fact, this quantification approach is a huge weakness of the work.

Author Response

Lines 135-136: “The assessment scale was established as a category scale ranging from “no perceived off-flavor" to “high intensity off-flavor". The category scale present how many “categories” or “different intensities positions”? This information should be provided. 

The description of the scale has been expanded according to the suggestion (now Lines 145146) as followingThe assessment scale was established as a 4-point category scale: “no perceived off-flavor", “low-intensity off-flavor", “moderate-intensity off-flavor", and “high-intensity off-flavor". 

Why only sulphur related compounds were used in training of the sensory panelIn lines 256 and 257, authors refer that “… Long chain fatty acid esters possess a characteristic “fatty” or “soapy” odor description;” These descriptors should also be evaluated in training sessions. 

The sensory panel and the study itself have focused on monitoring solely hydrogen sulfide related off-flavor because the overproduction of hydrogen sulfide is regarded as one of the main challenges in cider production management.  

The probability of the off-flavor formation related to elevated amounts of long chain fatty esters, on the other hand, in practice might not only depend on the concentration but the synergistic effects with other esters present in the sample. As already mentioned in the article, some studies even reported long chain fatty acid esters to be desirable in the wine fermentations. There are risks of off-flavor formation and they are acknowledged in the manuscript, but they are not an immediate challenge as hydrogen sulfide off-flavor management is and, hence, were not included in the sensory assessment. 

Lines 276-282: Some information of this sentences should be removed and placed in Materials and Methods section. Moreover, the trademark and supplier of the lead acetate paper strips should be provided. Authors should explain why they didn't use a more accurate and reliable method in order to quantify hydrogen sulfide. In fact, this quantification approach is a huge weakness of the work. 

Arequested, a couple of sentences were moved to the Methods section to Lines 98–102. 

The trademark and supplier of the lead acetate paper strips are provided in Lines 94–96 as follows: In the headspace of each vessel, a piece of lead acetate paper strip (Whatman; GE Healthcare, Chicago, IL, USA) was placed to monitor hydrogen sulfide formation.” 

The method for hydrogen sulfide formation assessment was chosen primarily based on the availability and simplicity of execution. The lead paper strip allowed us to monitor the hydrogen sulfide formation directly in the headspace during fermentation when yeasts are the most active. The method is indeed not very precise; however, it still allows us to assess if any drastic changes (e.g., a switch from no reaction on the strip to an intense reaction) occur when fermentation parameters are changed in any way (temperature, nutritional supplementation, etc.). As the results show, the minor changes were not detectable sensorially anyway, so the chosen method still provided us with enough information to tie it with an impact on the consumer level. 

Reviewer 2 Report

This paper entitled "The effect of apple juice clarification and concentration on cider fermentation and properties of the final products" focus on the application of freshly pressed and concentrate apple juices as substrate for cider fermentation. This paper is easy to read and I liked it. Literature used have a scientific soundness and ii is well written.

In my opinion the word "clarification" in the title should be deleted. It could be used as keyword. This word makes de work a little confused. Moreover, authors should discuss a little more about the differences in the fermentation preformances carried out by S.bayanus and S. cerevisiae. The behaviour of both yeast subspecies are usually different according to the literature.

Author Response

In my opinion the word "clarification" in the title should be deleted. It could be used as keyword. This word makes the work a little confused. 

The title of the article was changed to “The effect of apple juice concentration on cider fermentation and properties of the final product” according to the suggestion. 

 Moreover, authors should discuss a little more about the differences in the fermentation performances carried out by S.bayanus and S. cerevisiae. The behaviour of both yeast subspecies are usually different according to the literature. 

Due to no visible patterns of behavior in terms of fermentative activity, no generalized conclusions on S. cerevisiae vs S. bayanus could be made. 

Despite several reports on S. cerevisiae outperforming S. bayanus (e.gSchütz et al, 1995; Kishimoto et al, 1995), in our study, S. cerevisiae Y2 has shown the fermentative activity similar to S. bayanus Y1 and Y3. Therefore, in this work, we discussed fermentation activity of all six yeast strains and their reaction to the change in the fermentation environment separately. 

Shütz, M. & Gafner, J. (1995) Lower Fructose Uptake Capacity of Genetically Characterized Strains of Saccharomyces bayanus Compared to Strains of Saccharomyces cerevisiae: A Likely Cause of Reduced Alcoholic Fermentation Activity. Am J Enol Vitic 46, 175-180. 

Kishimoto, M. & Goto, S. (1995) Growth temperatures and electrophoretic karyotyping as tools for practical discrimination of Saccharomyces bayanus and Saccharomyces cerevisiaeJ Gen Appl Microbiol 41, 239-247. 

Reviewer 3 Report

The authors wrote a interesting and well-formed paper. The experiment is well-conducted and I have no major remarks. I did not, however, succeeded to follow the biplot (Figure 4). Could you please revise this part because I feel this could be a problem to understand for the broader audience. 

Author Response

I did not, however, succeeded to follow the biplot (Figure 4). Could you please revise this part because I feel this could be a problem to understand for the broader audience. 

Small explanation how to read biplots was added to the text (lines 259-262)PLS-DA method identifies the differences between the sample groups, which are shown as ellipses on the biplots with overlap indicating similarity. The characteristic differences between the groups are evaluated on an axis that goes through the origin point at [0, 0] towards the position of a volatile ester label. 

Additionally, sentence “The number in parenthesis shows variance captured by the component” was added to the Fig. 4 caption to explain an element shown on the plots. 

 Small grammatical corrections were made throughout the article e.g. article use, hyphens, commas, typos, etc.